# The Role of Sustainability in Telemedicine Services: The Case of the Greek National Telemedicine Network

**DOI:** 10.3390/healthcare13091046

**Published:** 2025-05-02

**Authors:** Fotios Rizos, Haralampos Karanikas, Angeliki Katsapi, Mariana Tsana, Vasileios Tsoukas, George Koukoulas, Dimitrios Drakopoulos, Aglaia Katsiroumpa, Petros Galanis

**Affiliations:** 1Euro-Mediterranean Institute of Quality and Safety in Healthcare, Avedis Donabedian, 10678 Athens, Greece; akatsapi@eiqsh.eu (A.K.); mtsana@eiqsh.eu (M.T.); 2Department of Business Administration, University of West Attica, 12241 Athens, Greece; 3Department of Computer Science and Biomedical Informatics, University of Thessaly, 35131 Lamia, Greece; karanikas@uth.gr (H.K.); vtsoukas@uth.gr (V.T.); 42nd Healthcare Region of Piraeus and Aegean Piraeus, 18233 Agios Ioannis Rentis, Greece; koukoulas@2dhr.gov.gr; 5Dextera Consulting, 15343 Agia Paraskevi, Greece; ddrako@dexteraconsulting.com; 6Clinical Epidemiology Laboratory, Faculty of Nursing, National and Kapodistrian University of Athens, 11527 Athens, Greece; aglaiakat@nurs.uoa.gr (A.K.); pegalan@nurs.uoa.gr (P.G.)

**Keywords:** e-health, telemedicine, sustainability, healthcare, Greek national telemedicine network

## Abstract

**Background:** Sustainability in healthcare has gained increasing importance due to its impact on environmental, financial, and social strategies, as well as on public health, and therefore, relevant policies and actions can also play a significant role in telemedicine services. The establishment of a sustainable telemedicine network at a country level is important to improve access to healthcare, reduce costs, increase convenience, and ensure the continuity of service delivery. However, there are significant environmental, social, technological, human, and governance challenges to meet the sustainability conditions for these networks. **Methods**: Thus, a narrative literature review was conducted to investigate the telemedicine implementation aspects and the sustainability dimensions in a unified approach and integrated strategy in order to develop a more resilient and equitable healthcare solution, ensuring its long-term integration into healthcare systems. **Results:** This paper aims to identify critical factors related to the proposed governance model for the National Telemedicine Network in Greece (EDIT) that influence sustainability requirements and interdisciplinary strategies to address relevant challenges. **Conclusions:** By examining these factors, the paper seeks to propose the fundamental pillars of a sustainable telemedicine framework and the methodology for developing a sustainability plan that will enhance EDIT’s capacities toward a sustainable and resilient operation of telemedicine as a standard practice within the Greek healthcare system.

## 1. Introduction

Concerns about climate change have intensified in recent years, aligning with a global shift toward sustainable development and a growing consumer preference for eco-friendly products and services [1,2,3]. Sustainability has become a key priority for organizations across both the public and private sectors. Recently, it has gained notable attention in the healthcare industry, particularly within hospitals [4,5,6], which serve a vital role in society [7,8]. As providers of essential medical services, hospitals bear the responsibility of ensuring high-quality healthcare for patients [4,9,10], especially for remote areas such as distant islands or other rural and mountainous areas that are difficult to access.

The impact of healthcare activities on environmental degradation and the environmental conditions’ critical role in public health have intensified discussions to enforce sustainable environmental and social strategy, worldwide [11,12]. Healthcare organizations (HCOs) must navigate the challenges of resource management, cost control, technological advancements, and the integration of environmentally sustainable practices that balance social, economic, and ecological considerations [13,14,15].

Hospitals and healthcare organizations are major consumers of essential resources, including energy, water, and raw materials. The healthcare sector is responsible for an estimated 4–5% of global greenhouse gas emissions, underscoring its pivotal role in promoting sustainability, particularly from an environmental perspective [16]. Significant reductions in CO_2_ emissions are expected over time while also enhancing patient care, increasing staff satisfaction, and achieving cost efficiencies [7]. Additionally, the operations of healthcare organizations contribute to advancing the 17 Sustainable Development Goals (SDGs), particularly in areas such as good health and well-being, gender equality, clean water and sanitation, and access to affordable, clean energy [17].

The need for a shift to circularity and sustainability strategies is recognized and addressed by, among others, the Circular Economy Action Plans introduced by the European Commission [18] and the United Nation’s Sustainable Development Goal (SDG), the EU Green Deal (reach zero GHG emissions and carbon neutrality and protect human health by 2050) [19,20]. Over the years, different strategies to reduce or even minimize the environmental impact of healthcare have been suggested toward waste management, energy–water consumption, GHG emissions, etc. [21,22,23,24]. The adoption of telemedicine services seemed a proper approach and strategy, especially when it comes to the reduction in GHG emissions in the healthcare sector [25,26,27].

Telemedicine has positively impacted global healthcare and the environment in recent years [5]. In order to comply with the constantly growing demands for healthcare resources without exacerbating climate change [28], the provision of healthcare services must be built on sustainable and low-carbon systems and work models [29]. Sustainability in telemedicine refers to the ongoing ability of digital health systems to provide fair and high-quality healthcare services across diverse regions and populations while taking into consideration environmental, social, governance, and ethical considerations, ensuring lasting effectiveness in clinical, economic, and organizational terms.

It is proven that information and communication technology (ICT) is a solution to reduce the carbon footprint [30]. Moreover, the proper adoption and implementation of telemedicine services would be an answer to greener healthcare systems and for the protection and preservation of the environment [31]. In addition, several measures to reduce the carbon footprint of the healthcare sector have been identified in the framework of telemedicine [24], such as the use of reusable devices [32], reducing waste [33], and one-stop diagnostic solutions [34].

Based on the reviewed literature, it is acknowledged that in the implementation of telemedicine services, given the underlying challenges and the outcoming benefits, a framework establishing sustainability principles must be considered. Thus, in the present work, we attempt a first approach to investigate and propose a proper sustainability framework in the governance models of telehealth. Consequently, the main research question that follows our paper is as follows:

RQ: “How can the Greek national telemedicine network apply sustainability principles under the proposed governance model and what is the methodology to develop a suitable sustainability plan?”

Thus, we performed a narrative literature review in order to investigate the telemedicine implementation aspects and the sustainability dimensions in a unified approach and integrated strategy. The main research objectives that this paper is trying to answer, but are not limited to, are as follows:What is the current theoretical background that arises from the literature review regarding telemedicine and sustainability?What are the potential challenges?Can a mutual conceptual framework be created with respect to the characteristics of the current governance model of telemedicine?

This research is structured as follows: In Section 2, the theoretical background and a brief presentation of the methodology of this paper are presented along with the challenges and expected opportunities of telemedicine services. In addition, the structure and components of the national telemedicine network in Greece (EDIT) are briefly presented.

In Section 3, the findings of our research, also based on the author’s experience, are presented, including the proposed fundamental pillars of a sustainable telemedicine governance framework, and in addition, the proposed methodology for developing a sustainability plan for telemedicine services in Greece with the respective steps to be followed.

Finally, Section 4 presents the concluding comments and remarks, along with the research gaps, the potential need for future academic research development, and the implications of this paper.

## 2. Theoretical Background

### 2.1. Methodology of Our Research

Our purpose was to conduct comprehensive research, and thus, the systematic literature review’s guidelines were followed according to Yu and Watson [35] and Tranfield and Smart [36] and our methodology is presented is Figure 1. The first step, after defining the research questions and proposed objectives, was to develop the relevant review protocols and methods with all the necessary inclusion and exclusion criteria. A committee of experts was appointed, consisting of the following review authors/experts: three in the field of telemedicine, one in the field of sustainability in the healthcare sector, three healthcare professionals, and two academics with experience in the aforementioned fields as coordinators. The review authors performed a parallel independent assessment of the articles and reports, and in case of a disagreement or dispute, the coordinator was responsible for giving a solution. For our search, we used the online databases of Google Scholar and PubMed, which contain journal articles as well as “grey literature”, such as conference proceedings and reports, and reviewed the first 19 pages of search results from the last two decades. The search was performed using the following terms, keywords, abbreviations, and combinations of e-health, telemedicine, telehealth, sustainability in telemedicine, digital health, sustainability in healthcare, Greek national telemedicine network, and ESG in healthcare, and only publications written in English.

At the first stage, we identified and selected the initial group of 1181 studies that were screened through their titles, the context of the abstracts (for papers), and the table of contents (for reports and non-academic publications) in order to investigate the relativity of the papers based on the research question and relative objectives. The next step was to select the papers, publications, and reports that were close enough to answer them. A total of 197 studies were deemed relevant, and we obtained the full-text article for quality assessment. Then, we skimmed through the full-text articles to further evaluate the quality and eligibility of the studies. After careful review, a total of 89 studies were excluded, and altogether, we included a total of 108 studies in this research that meet our inclusion and exclusion criteria.

After analyzing and synthesizing the existing information from the selected studies, we codified it into specific concepts, such as the challenges and benefits of telemedicine with regard to sustainability aspects, and then presented our findings. Afterward, we concluded with the suggestion of a conceptual model for the governance model of the Greek national telemedicine network. The implications of this research are quite significant for all healthcare actors and stakeholders (e.g., patients, healthcare providers, health institutions, academia, healthcare experts and executives, governments, etc.) because it presents an updated version of the current operating telemedicine system in Greek toward the sustainability principles and the modern philosophy of governance in digital healthcare services.

### 2.2. Challenges and Expected Positive Outcomes of Telemedicine

In the field of telemedicine, a lot of crucial challenges have been identified by different researches [37,38]. Pan et al. [38] refer to behavioral and operational aspects and the unwillingness to change during the implementation of telemedicine services, especially on the part of clinicians and non-clinicians in terms of smart healthcare systems’ adoption [39]. Moreover, Rubbio et al. [40] and Martínez-Caro et al [41] highlight the adoption difficulties and resiliency to change when it comes to implementing digital health technology platforms. Furthermore, Cobelli et al. [42] have identified ethical challenges such as privacy and security issues, the lack of autonomy, and other legal implications and liabilities [43], while other researchers present the high technological costs and the technical aptness of the staff [44,45]. From the patient’s perspective, the lack of adaptation to the new technological tools [46,47], the limited broadband access [48], and the language barriers [48], the lack of trust and acceptance, as well as the digital literacy, are major challenges in the adoption of telemedicine services [45,48,49], in addition to other critical financial and infrastructural deficiencies [48,49,50,51,52].

Nevertheless, the positive impact of telemedicine services has been proven over the last decades and has the potential to greatly contribute to sustainability and support the development of an eco-friendly and socially based healthcare system [48]. Adopting telemedicine not only benefits the environment because of the reduction in transportation but also enhances access to care and improves patient specialized management through distance consultations, timely interventions, and individualized treatment options, ultimately improving health outcomes. By implementing these practices, healthcare practitioners and telemedicine platforms can greatly contribute to the battle against climate change, the preservation of natural resources, and the establishment of a more sustainable future [48]. Telemedicine is considered a beneficial and suitable approach toward sustainable healthcare [37,53]. In fact, telemedicine also has multiple benefits that should be mentioned with regard to society and the health systems [54]. Research has shown that telemedicine services can improve the overall performance of the healthcare systems [55], enhance patients’ satisfaction [56], and increase the quality of services [57] by reducing the patients’ waiting time [58]. In the same direction, telehealth services mitigate geographical barriers and increase access to healthcare services [30], reduce the emissions from patients’ medical travel [58], reduce the overall healthcare emissions [59], minimize transportation and traveling time and costs [28,60], and contribute to a more efficient healthcare system because of the reduction in unnecessary hospital admissions [28,61]. The patient-centered experience due to the real-time communication with healthcare providers through videoconferences [62] enables continuous documentation of medical data [62] and provides easier follow-up and monitoring processes by healthcare professionals [62]. Moreover, it is imperative that competencies in ICT tools, virtual platforms, and applications must be enhanced both from the healthcare providers’ and the end-users’ sides [63,64]. The introduction and implementation of telemedicine services with their respective supportive systems and tools should be a responsibility [37] and, in some cases, the obligation of healthcare policy decision-makers at the micro level (e.g., in healthcare entities) and at the macro level (e.g., healthcare ministry) [42]. In the next section, a methodological approach to the governance of telemedicine networks in Greece, with sustainability as the main purpose, is presented.

### 2.3. The National Telemedicine Network in Greece (EDIT)

In response to ongoing challenges, the Greek Ministry of Health has been expanding the National Telemedicine Network to enhance healthcare services, particularly in remote islands, mountainous rural regions, and other inaccessible locations. This initiative aligns with Greece’s constitutional commitment to ensuring equal healthcare access for all citizens, regardless of their place of residence. Currently, the Greek national telemedicine network (EDIT) includes [65,66] the following:Sixty-six Patient-Doctor Telemedicine Stations (PDTS), situated in hospitals, health centers, and multipurpose regional clinics.Twenty-one Consultant Telemedicine Stations (CTS), located across 12 regional and tertiary hospitals in the 2nd Greek Health Region (HR), as well as in the National Emergency Centre (NEC).Over 170 Home Care Stations (HCS) installed in patient homes or social care facilities within the 2nd Greek HR’s jurisdiction.

To further enhance telemedicine capabilities, the Greek Ministry of Health is implementing significant upgrades, including additional infrastructure and subscription-based services. The planned improvements include the following:Three hundred and fifty-five new Patient–Doctor Telemedicine Stations (PDTS) to be deployed in selected healthcare facilities across the country.Thirty-five additional Consultant Telemedicine Stations (CTS), strategically placed based on healthcare facility capacity and operational needs.Five Telemedicine Training Centers, equipped with both CTS and PDTS technology, to train healthcare professionals in university hospitals nationwide.Three thousand Home Monitoring Systems (HCS), integrated with EDIT and its supporting software, to improve remote patient monitoring.Three new regional Control Centers and a centralized Command and Control Center at the Ministry of Health to oversee telemedicine operations.

Between 2016 and 2023, Greece witnessed a substantial increase in teleconsultation services, particularly in mental health [65,66]. Telepsychiatry sessions emerged as the most common form of telemedicine consultation, followed by pediatric telepsychiatry, diabetology consultations, and chronic disease management services. The continuous evolution of telemedicine in Greece demonstrates its vital role in improving healthcare accessibility, particularly for individuals living in isolated or underserved areas who face barriers to traditional medical services. To maximize the effectiveness of EDIT, it is essential to implement clear regulations and standards that uphold quality care and ensure optimal clinical outcomes [65,66].

Expanding telemedicine, particularly in Greece’s remote and mountainous regions, for prompt response to cover the needs of chronic patients or for health problems consultations represents a strategic and necessary approach to fulfilling the constitutional obligation of universal healthcare access. But, in order to be sustainable and take into consideration the necessary aspects of the ever-changing environment, a proper framework and methodology should exist to verify the continuous existence and improvement in terms of meeting the needs and expectations of the end-users and the stakeholders [65,66].

## 3. Results

Our study is based on literature review and on experience in the telemedicine field. The purpose of this section is to present a proper framework based on the principles and philosophy of sustainability that will support and improve the governance model of EDIT so as to meet the challenges and risks that organizations are currently facing.

### 3.1. Fundamental Pillars of a Proper Sustainable Telemedicine Framework

Telemedicine services in the healthcare sector are using various tools and applications, such as information and communication technology (ICT) platforms, to deliver the necessary and expected healthcare support and services [37,67,68,69]. It is interesting to mention that Chauhan et al. [37] have identified 37 key success factors concerning the delivery of the proper telemedicine services, with under seven major dimensions (criteria): social, environmental, economic, technological, legal and regulatory, ethical, and organizational. In this study, and more specifically in Figure 2, we present a conceptual model where the fundamental pillars of a proper sustainable telemedicine framework for the governance of the Greek national telemedicine network should consist of the following: environment (E), social (S), governance (G), economic (E), digital transformation and innovation (DTI), the legal framework (L), quality assurance (QA), and finally, the ethical considerations (EC).

Our research indicates that these pillars are complementary, mutually dependent, and supportive and should be considered as a holistic model in terms of successful implementation.

Each pillar consists of specific factors to be identified, considered, measured, evaluated, monitored, and revised. Appendix A presents the table of the proposed factors per pillar. In addition, this study tries to make a first attempt to show the correlation between the above-mentioned factors based on the authors’ empirical research. In Figure 3, it is clearly demonstrated that all factors are interlinked and have a mutual influence in an undeniable manner. In any case, our findings are based on empirical application and experience, thus, not conclusive.

Further research needs to be carried out due to the promising findings presented in this paper, and the work on the remaining issues is continuing and will be presented in future papers.

### 3.2. The Proposed Methodology for Developing a Sustainability Plan in Telemedicine Services in Greece

Sustainability has become vital for the protection of society and the environment. Although telemedicine is facing challenges, it is still considered an appropriate method to deliver and provide qualitative healthcare services in remote areas or even in developing countries where there are persistent shortages of specialization and healthcare resources [70]. Various studies have proven that the existence of a proper framework should be available in telemedicine implementation, and many researchers have presented different methodologies [70,71,72,73,74], but the interesting fact is that sustainability does not appear as an aspect or action dimension. In Greece, a governance framework has been recently proposed for the national telemedicine network to ensure alignment with regulatory requirements, quality dimensions, stakeholder expectations, and compliance with ethical and technical standards, while promoting innovation and excellence in the practice of telemedicine [66]. One of the key principles of the governance model is sustainability so that the network can adapt to the environmental, social, technological, financial, or any other needs and developments that affect the healthcare ecosystem.

This study suggests the key factors that concern the sustainability of telemedicine and the relevant methodology for their adoption consist of the following steps and relevant targets (Figure 4):

Involvement of the necessary authorities: In the proposed governance framework, the Greek Ministry of Health is the authority that sets the strategic plan. This includes the allocation of necessary resources and the needed directions, policies, and guidelines for the implementation of telemedicine services at the seven healthcare regions by the coordination of the Operational Centers of the regions.

The Coordination Directorate is accountable for the operational management. This is supported by three newly established units: the Business Operations Department, the Technological Infrastructure and Innovation Department, and the Communication, Publicity, and Digital Media Department. These departments are responsible for coordinating, guiding, and assisting the Operational Centers located in the regions to ensure seamless daily operations through the application of a well-defined framework of actors and standardized procedures.

Appointment of the appropriate committees: In this governance model, three significant committees are defined that will operate under the Greek Ministry of Health. The first one is the Coordination Committee which will be responsible for the overall application of the Greek national telemedicine program and will be deemed a high-level advisory and decision-making body. The Quality Committee will be responsible for implementing safety guidelines and telemedicine protocols, enhancing the quality of the telemedicine services toward continuous quality improvements. The Technological Standards Committee will be responsible for the establishment of technical protocols and respective standards related to the infrastructure, interoperability, and security. Finally, a Telehealth User Advisory Committee is proposed to ensure that patients and community members have a voice in the design, implementation, and assessment of the telemedicine program.

Definition of purpose and scope: The first step is to establish a clear understanding of the key elements, such as the need, purpose, and goal of a conceptual model in telemedicine services toward sustainability. In addition, it is necessary to define a summary of the structure, the type of telemedicine services to be included, the key stakeholders and other interested parties, the technological aspects, the requirements’ framework for quality and safety assurance, the basic governance principles including ethical and legal considerations, and the boundaries and limitations within which it operates [66].

Evaluation of the current situation: A healthcare entity should perform a self-assessment for identifying the status of the sustainability approach of its operations, especially at the Top Management level. Specifically, the healthcare entity should identify critical aspects such as the level of knowledge regarding sustainability and the existence of already implemented policies, Standard Operating Processes and activities for sustainability.

Conduct a materiality assessment: Oll et al. [75], Gerwanski et al. [76], and Karagiannis et al. [77] claim that materiality can enhance quality, transparency, and integrity and is considered a key tool for identifying the sustainability issues in corporate strategy and risk management processes [78]. To this end, a proper materiality analysis should be performed to specify the criteria and principles of the international sustainability standards that are important for the major stakeholders of the network [75,79,80]. Studies claim the importance of materiality as a fundamental concept by highlighting how materiality determines the importance of issues [75,81,82] and is linked to sustainability issues’ value relevance [75,83,84,85,86,87,88,89,90]. In addition, through the materiality process, the risks and opportunities and their impact on the fundamental pillars of the sustainable telemedicine framework are identified. That is considered an important tool for strategic planning for the decision-making process, with application in telemedicine services [84,85].

Develop a proper transition plan: Based on the findings of the self-assessment and the materiality assessment, a gap analysis should be performed in order to identify, analyze, and perform the proper actions toward sustainability requirements and then identify areas for improvement. It is vital that the applied policies, SOPs, processes, targets, and KPIs are aligned with sustainability aspects in order to create the sustainability transition action plan. The sustainability plan should also include the resources needed for the system to be designed, individual stakeholder capabilities, system characteristics aligned to the organizational strategy, and the applicable standards. In addition, the impact–risk and opportunities (IRO) should be taken into account, and sustainability alternatives should be defined in prior [70,91,92].

Implementation of sustainability actions and monitoring process: After the sustainability transition plan is defined and approved, the implementation phase begins with specific timeframes and accurate planning to improve the sustainability aspect of the telemedicine network. This process is vital because it requires a lot of resources and changes to the already established philosophy. Another essential element is the monitoring of the whole process and the implementation of sustainability initiatives and the respective progress in terms of continuous improvement. More specifically, regularly tracking progress toward sustainability goals using the identified metrics should be performed through internal audits to ensure adherence to the reported goals and identify areas for improvement. Moreover, maintaining open channels for stakeholder feedback is critical to ensure that the hospital’s sustainability practices evolve in line with stakeholder expectations and global trends.

## 4. Discussion

The progression and long-term viability of telemedicine in Greece have been significantly influenced by both European Union digital health directives and the country’s own strategic health plans. Initial steps toward adoption were facilitated through EU-backed initiatives like TALOS and FEST during the 1990s, which allowed island-based clinics to transmit ECG data to mainland hospitals. Later, projects such as Hygeia-Net, launched in 1998, reflected Greece’s ongoing efforts to expand healthcare access in geographically isolated areas. Between 2006 and 2015, the national eHealth roadmap—developed in alignment with broader EU digital priorities—formally acknowledged the importance of telemedicine in reducing regional health disparities. Despite this, progress was constrained by challenges including insufficient legal frameworks, the absence of structured reimbursement systems, and disjointed digital infrastructure. A notable advancement came in 2011 with the creation of the National Telemedicine Network (EDIT), which has since expanded, especially with financial support from EU cohesion programs and post-pandemic recovery plans. Currently, EDIT plays a central role in delivering healthcare remotely, particularly to hard-to-reach populations. However, its continued success relies heavily on harmonizing with EU-level policies such as the European Health Data Space (EHDS) and on strategic investments in areas like system interoperability, healthcare provider training, and digital competence. Greece’s journey underscores the critical influence of transnational policy coordination on ensuring the durability and scalability of digital health systems.

Moreover, in other European countries like the Netherlands [93], Norway [94], Sweden [95], Estonia [96], Spain [97], and France [98], the existing telemedicine models are taking into account ESG considerations along with economic aspects and have very strong governance frameworks. In Greece, the EDIT network is still under development in terms of the governance structure. Implementing a national telemedicine network is very challenging, and its sustainability is significant. Requirements to be included and considered are the financing and resources [24,37,39] (e.g., procurement and contracting framework), technology and infrastructure [39,43,44] (e.g., reliable and secure connectivity, equipment), leadership and governance [46,47,48,49,50,51,52,53] (e.g., safety and quality standards), staff management and training [28,59,60,61,62,63,64], beneficiary management (patients and caregivers) [25,28,59,60,61,62,63,64] (e.g., accessibility), evaluation and research [65,66,67,68,69] (e.g., risk assessments and information utilization), and innovation [28,43,44,45,46,47] (e.g., integration of innovations). These elements are crucial in terms of sustainability for the successful development of a national telehealth system.

The healthcare sector needs to rely on a sustainable and resilient operational model, and the respective investments have become an important tool that concerns both public and private stakeholders seeking to achieve health-related goals [16,70,99,100,101,102], achieve financial sustainability, and meet the needs and expectations of the interested parties, especially for society in terms of social impact [103,104,105]. Applying a sustainable healthcare model that considers environmental, social, and governance (ESG) factors in its operations will ultimately address business challenges [16] and align with the Sustainable Development Goals (SDGs) [106,107,108,109]. ESG integration not only enhances the long-term viability of investments but also strengthens stakeholder trust and social license to operate [110,111,112,113].

## 5. Conclusions

According to our literature review, we found evidence and references to support the proposed fundamental pillars of the sustainable telemedicine framework. Gray [114] and Oliveira et al. [115] mention the importance of the environment and also the need for environmental impact assessment to ensure that the developed system works in harmony with the environment and also to have evidence so as to reduce the negative effects in terms of emission, waste, energy and water consumption, etc. Kituyi et al. [70], Chetty et al. [116], and Carcary [117] claim that telemedicine has a social sustainability component and highlight the importance of considering and involving the local community and society in general so as to equalize the healthcare system due to catholic and overall access from all patients [118].

Moreover, the governance aspect plays a crucial role in the adoption of telemedicine services [48,119,120] and can significantly influence their sustainability. The economic aspect is found to have a very strong connection with telemedicine services, mostly regarding economic policies on funding models and reimbursement strategies surrounding telemedicine and the reduction in costs [121]. Digital transformation can play a fundamental role in the provision and improvement of the whole telemedicine structure [122,123] and in achieving universal health coverage and health-related goals [124,124,125]. Furthermore, in the proposed governance framework for the national telemedicine network in Greece [65,66], the legal framework, quality assurance, and ethical considerations are deemed imperative concepts and elements for the successful and efficient implementation of telemedicine services in order to align with stakeholder expectations.

Due to limited resource availability, immediate and radical actions are not easy to implement. It is important to highlight that sustainability in telemedicine services presents several challenges that must be carefully addressed to ensure long-term viability and effectiveness. One major concern is the financial sustainability of telemedicine infrastructure, as the initial investment in technology, equipment, and network expansion can be costly, requiring ongoing funding for maintenance, updates, and training. Additionally, disparities in digital access and technological literacy, particularly in rural and underserved areas, can hinder widespread adoption and equitable service delivery. Regulatory and legal challenges, including data privacy, cybersecurity, and compliance with healthcare standards, also pose significant barriers to sustainable telemedicine implementation.

Furthermore, the integration of telemedicine into existing healthcare systems requires careful planning to avoid disruptions in service continuity, ensure interoperability between platforms, and maintain high-quality patient care. Moreover, sustainability in telemedicine services should extend beyond financial and technological challenges to include environmental, social, and governance (ESG) concerns, as well as the impact of climate change, and the digitization of healthcare services is able to promote sustainable practices in the healthcare sector.

Finally, this paper aims to be a beacon for future research on sustainability in the provision of telemedicine services. The limitations of our paper are the outlined principles and research methodology that require further in-depth investigation based on a systematic literature review, and there may also be more factors that influence telemedicine than those investigated and theoretically introduced. Although various factors have been demonstrated per category, more field research needs to be carried out. So, the intent is to pilot-test and validate the proposed governance model empirically, based on the factors and KPIs for the fundamental pillars of the sustainable telemedicine framework, by applying it to selected case studies in EDIT, evaluating their structures, stakeholder engagement mechanisms, and policy frameworks. The analysis will examine how these components support the long-term sustainability of telemedicine services in Greece, employing both qualitative evaluation methods and quantitative performance indicators.

## Figures and Tables

**Figure 1 healthcare-13-01046-f001:**
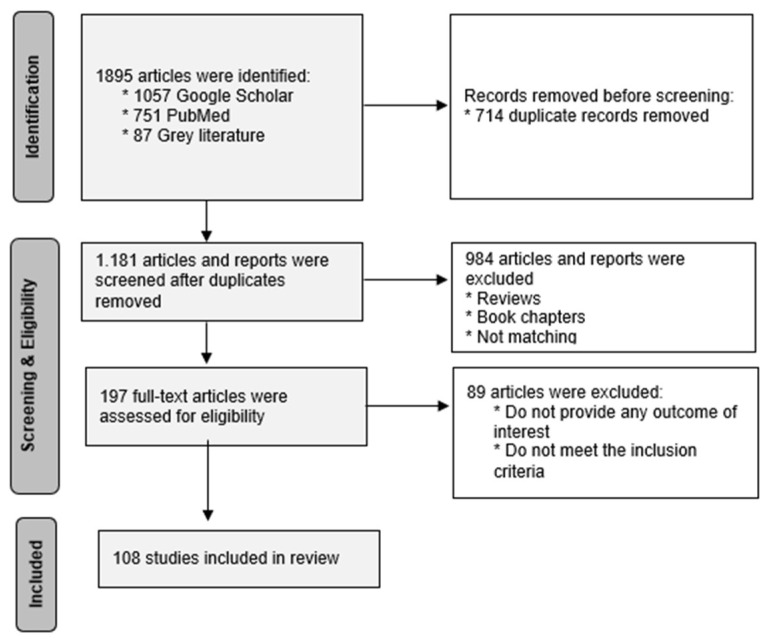
Methodology based on PRISMA principles. Source: Derived from our research.

**Figure 2 healthcare-13-01046-f002:**
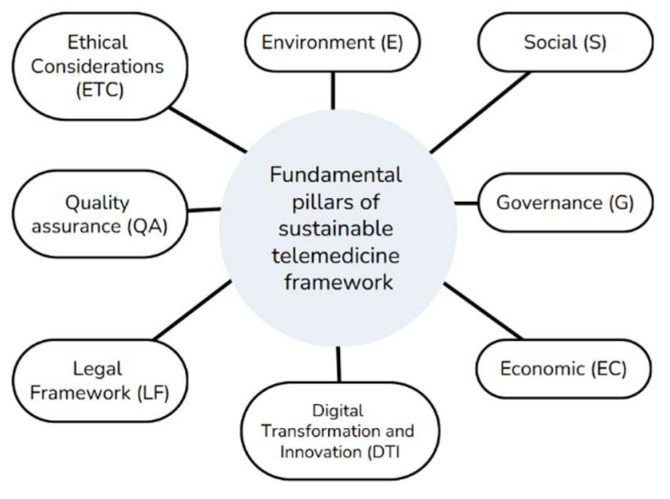
Fundamental pillars of sustainable telemedicine framework Source: Derived from our research.

**Figure 3 healthcare-13-01046-f003:**
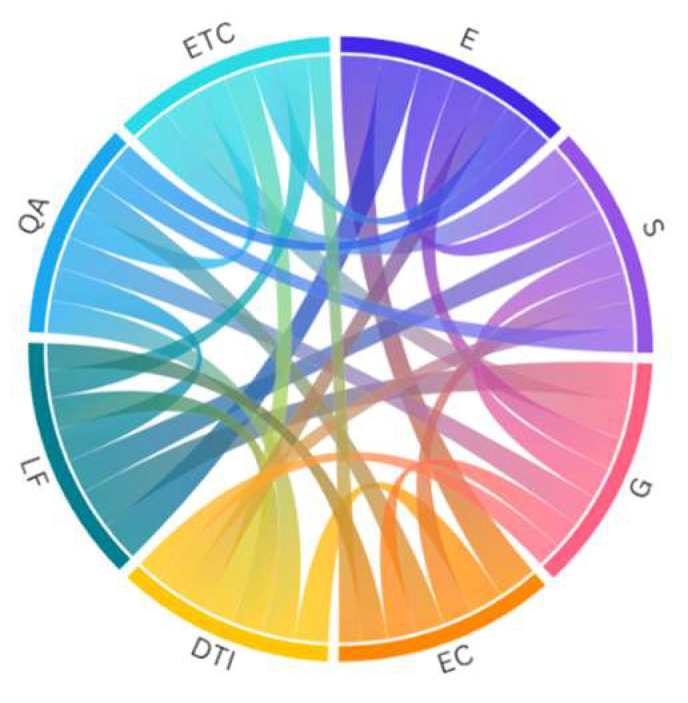
Correlation of the fundamental pillars of sustainable telemedicine framework Source: Derived from our research.

**Figure 4 healthcare-13-01046-f004:**
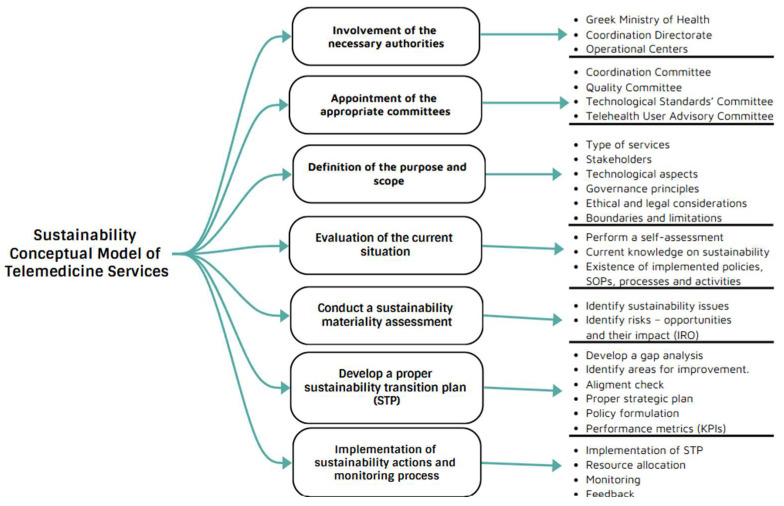
Proposed steps of the methodology for developing a sustainability plan in telemedicine services in Greece Source: Derived from our research.

## Data Availability

Data are contained within the paper.

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
