# Peer review of "The Role of Sustainability in Telemedicine Services: The Case of the Greek National Telemedicine Network"

_healthcare, 2025, doi:10.3390/healthcare13091046_

Round 1

Reviewer 1 Report

Comments and Suggestions for Authors

General Overview

Your manuscript addresses an important and timely topic on sustainability in telemedicine, specifically within the Greek National Telemedicine Network (EDIT). The paper’s ambition to integrate environmental, social, economic, and governance considerations into a cohesive telemedicine sustainability framework is valuable. Overall, the manuscript has a clear direction and purpose. See my  specific, section-by-section feedback followed by suggestions for improvement.

Introduction

    • While climate change and environmental impact are core motivations, consider more direct references to Greece’s unique geography (islands, mountainous regions) earlier in the introduction. That immediately underscores why telemedicine is so critical and how sustainability ties into bridging access gaps.
    • The stated research question (lines ~85-86) is valuable, but the introduction would benefit from a succinct summary of the network’s historical progression (e.g., from pilot e-Health projects to a national telemedicine platform) and how sustainability fits into that evolution.
    • Consider providing a concise operational definition of “sustainability in telemedicine” as you use it. For instance, clarify how economic, social, environmental, and technological pillars convergence would help ensure readers understand your conceptual approach.

Literature Review

  • There is a need for stronger organization by pillar or theme. The discussion often intersperses environmental, social, legal, and technological issues. You might re-organize around the four or five main pillars (e.g., environment, social, economics, technology, governance). This would reinforce your conceptual model and demonstrate how each pillar is validated by existing literature.
  • Your references to Greece’s constitutional commitment and mountainous/island geography are strong but somewhat scattered. A more concentrated subsection summarizing Greek healthcare constraints (economic crises, diaspora, island remoteness) would clarify why telemedicine and its sustainability are so pivotal.
  • Consider including a brief mention of how EU digital health policies (or national eHealth strategies) historically impacted telemedicine adoption in Greece. This link is essential to show how the network’s sustainability also depends on broader policy alignment.

Methods

Clarify how you refined or combined your keywords (e.g., Boolean operators or synonyms for “telemedicine + sustainability + Greece”) would improve reproducibility.

    • Although you mention “inclusion and exclusion criteria,” you do not detail them thoroughly. Briefly state the main criteria (e.g., date range, telemedicine in healthcare, sustainability focus).
    • The text indicates you coded challenges and benefits but does not elaborate on how that coding was performed or validated. A short paragraph on the thematic analysis or coding categories would add rigor.

Results

    • The paper includes strong generic pillars but less explicit reference to actual usage data, number of teleconsultations, or cost-effectiveness from the Greek network. Even anecdotal examples from EDIT data (without requiring new data collection) could illustrate how these pillars mitigate real problems. Otherwise, the framework appears somewhat abstract.
    • You propose important steps (e.g., self-assessment, materiality analysis). Clarify how you see these steps integrated in the actual day-to-day governance of EDIT. For instance, who exactly performs the materiality analysis? The mention of committees is good, but linking each step to a responsible entity in the governance chain would make it more concrete.

Discussion

    • This section should reference or contrast with other national telemedicine sustainability models (e.g., from another EU country). A short paragraph highlighting differences and parallels could enrich the discussion and show the Greek approach’s uniqueness.
    • The core pillars might create synergy or conflict (e.g., cost reduction vs. advanced digital tools that require heavy initial investments). In the discussion, exploring these trade-offs or synergy points would demonstrate a deeper grasp of real-world complexities.

Conclusion

    • The conclusion could provide more direct recommendations for policy-makers—e.g., specific incentives or funding models to preserve the network’s operational resilience beyond pilot phases.
    • Reiterate the major steps from your methodology in a succinct bullet point or short summary form, so the last paragraph underscores exactly what next steps look like for Greece.
Comments on the Quality of English Language

The English could be improved to more clearly express the research.

Author Response

Dear esteemed Reviewer.

Please see the attachment in response to your comments. 

We remain at your disposal for any questions or additional actions to be taken.

With respect. 

Reviewer 2 Report

Comments and Suggestions for Authors
  • please make sure to have concise and short sentences since there are many long, complex sentences that obscure meaning.
  • please add table summarizing the governance structure and responsibilities to clarify it more.
  • How the KPIs were selected. is it validated?
  • Why did you select  narrative, systematic and justify why this approach was appropriate comparing to scoping and systematic 
  •  what is “gray literature” in this study
  • please add why you used the “first 19 pages” in Google Scholar.??
  •  How the quality of the 108 studies was assessed
  • How data were extracted ( thematic coding, matrix framework) by whom, and if there were any disagreements between reviewers and how were handeled.
  • Add a table for the Frequency of each theme/pillar in the literature and the Number or percentage of studies supporting each pillar

    •  

Author Response

(The authors gave the same response as above.)

Round 2

Reviewer 1 Report

Comments and Suggestions for Authors

Your revisions successfully address most prior concerns: methodology is now transparent, the governance framework is better anchored in Greek/EU policy, and language readability has improved. Before final acceptance, please:

  1. Correct two residual typos (“remoted” → “remote” line 42, “Ethonia” → “Estonia” in Line 400).
  2. Add a brief sentence in the Conclusion stating how you intend to test or apply the framework empirically.

Author Response

Dear esteemed reviewer.

I would like to thank you once again for your fruitful comments and guidance. Regarding the second round, we have corrected the typos as you mentioned and we have added a short sentence (lines 466-472) stating how you intend to test or apply the framework empirically. Please find attached the revised version of the manuscript (with track changes and red color) We hope that this is sufficient. 

If you have any other comments, please let us know.

With respect.

Thank you again for your time and effort which is greatly appreciated. 

Reviewer 2 Report

Comments and Suggestions for Authors

accepted 

no comments 

Author Response

Dear esteemed reviewer.

I would like to thank you once again for your fruitful comments and guidance. Please find attached the revised version of the manuscript (with track changes and red color) 

With respect.